# Mediastinal Metastasis Isolated in Ovarian Cancer: A Systematic Review

**DOI:** 10.3390/life14091098

**Published:** 2024-08-31

**Authors:** Victoria Psomiadou, Alexandros Fotiou, Christos Iavazzo

**Affiliations:** 1Metaxa Memorial Cancer Hospital, 51 Botassi Str., 18537 Piraeus, Greece; christosiavazzo@hotmail.com; 23rd Department of Obstetrics and Gynecology, Athens Medical School, National and Kapodistrian University, Attikon Hospital, 12462 Athens, Greece; alexandrosfotiou92@gmail.com

**Keywords:** ovarian cancer, isolated mediastinal metastases, ovarian metastases, surgical debulking, cytoreduction

## Abstract

Background: Isolated mediastinal metastases from ovarian carcinoma are considered exceptional. Since such metastases are considered advanced stage disease, systemic therapy is the indicated therapeutic approach; however, some articles report that surgical excision is also feasible. Methods: We reviewed the English-language literature to detect cases of isolated mediastinal ovarian cancer metastases and present the management applied as well as their outcomes. Results: From 1998 to 2022, 15 such cases have been reported, with 4 of those cases being primary ovarian cancer presentation and 11 being ovarian cancer recurrence. The histology of the tumor was serious in all of the cases. Regarding the management of cancer, various methods were applied. In total, 11 of the patients underwent a surgical resection of the mediastinal metastasis, 2 received systemic therapy, 1 received a combination of palliative chemotherapy and radiation and the last patient was treated with laser debulking and radiotherapy. The mean reported follow-up was 11 months. Conclusions: Solitary mediastinal metastasis from ovarian cancer is very rare; physicians should pay close attention when routinely evaluating thoracic scans from patients with ovarian malignancy as well as individualizing the management in such patients, since surgical resection can also be performed. However, definitive conclusions cannot be drawn from the small number of case reports available.

## 1. Introduction

The rate of mediastinal dissemination in ovarian cancer (OC) is estimated at only 2.3%, which is mainly believed to occur via locoregional spread, following a specific pattern where the primary tumor spreads in the mediastinal space, located between the pleural cavities, which enclose the lungs (Figure 1) via intrathoracic lymph nodes metastases. Usually, isolated mediastinal metastases are rare and appear late in the course of the disease and when they are present, they are usually a manifestation of widely disseminated disease [1]. However, the reported cases of mediastinal involvement in ovarian cancer patients are increasing, which may be due to the progress made regarding the imaging techniques and the chemotherapeutic agents as well as the overall longer survival rate of ovarian cancer patients. Recently, thoracic cytoreduction has been proposed in such cases either with the intent to either cure or to extend the survival rates [2]. However, the estimated median survival time after extra-abdominal metastases in OC patients is unclear, since the existing literature on these cases is limited to case reports and no randomised trials exist to strongly support surgical approaches against systemic chemotherapy in these scenarios. Nevertheless, their detection and distinction from metastases from different primary sites are of huge clinical importance because the treatment and prognosis diverge significantly, although the main treatment so far as for any general metastatic cancer consists of palliative chemotherapy without curative intent. This underscores the importance for clinicians to contemplate mediastinal metastasis during OC patient follow-ups. Successful cytoreduction surgery (CRS) for this less frequent metastatic site has been reported with promising outcomes but due to its low incidence, little data are available [3]. Metastasis to the mediastinum from OC can lead to severe complications due to direct pericardial invasion. In instances of a solitary mediastinal metastasis, surgery could be the most effective treatment option. However, a substantial number of cases need to be gathered to determine the best management strategy, and a multidisciplinary approach and individualized treatment are always crucial in managing rare gynecological cancer cases, since the importance of personalized and innovative treatment strategies in rare metastatic conditions has been emphasized in many studies, suggesting that radiotherapy or neoadjuvant chemotherapy could also be considered [4,5,6]. In the present review, we aim to investigate the possible role of secondary cytoreduction in patients with isolated mediastinal ovarian cancer disease to identify related needs, explore feasibility and reveal outcomes associated with different treatment approaches.

## 2. Materials and Methods

### 2.1. Data Sources

We searched PubMed, Google Scholar and Cochrane Library databases from inception until August 2024. To minimise the possibility of report losses, we also performed snow-balling of the references of articles that we retrieved. Our search strategy included the text word (isolated mediastinal) AND (metastasis OR involvement) AND (ovarian cancer OR tubal OR peritoneal) and is schematically presented in the PRISMA flow diagram (Figure 2).

### 2.2. Study Selection Criteria

The selection of studies was conducted in three consecutive stages. After checking for duplicate publications, two authors (VP and AF) screened the titles and abstracts of electronically retrieved articles to determine if they were eligible for inclusion. The decision was finalized after retrieving and reviewing the full texts of articles that were considered to be relevant to the topic. Any discrepancies that arose among the two authors during these steps were resolved by consensus of all authors.

### 2.3. Selected Studies

Eligibility criteria were predetermined. No language restrictions were applied during the electronic search but we excluded all articles that were written in languages other than English. All studies that investigated histological or radiological diagnosis of isolated mediastinal metastasis of an ovarian, tubal or peritoneal carcinoma as well as the outcomes of various treatments that were implemented in these patients were included in the present systematic review, provided that these recurrences were isolated and no other organs were involved in disease relapse. The stage of the disease at primary diagnosis was not considered a criterion for inclusion, nor the extent of the follow-up period; however, differences in baseline characteristics of patients were recorded and tabulated when available. Conference abstracts were also considered as eligible and tabulated. Editorials, comments and reviews were not included in the present systematic review.

### 2.4. Literature Review

The literature search revealed 93 potentially relevant articles of which the abstracts and titles were screened. In total, 19 articles were included for full-text evaluation, of which 6 were excluded because they presented cases with multiple metastases or extended disease, and 3 titles referred to radiological criteria to distinguish the metastatic mediastinal disease from other entities. Studies on mediastinal metastasis in OC patients were either retrospective case reports or case series on patients with OC. Table 1 summarizes the patients’ and the disease characteristics as well as the management applied and the follow-up of reported mediastinal metastasis (MM) in these studies. Finally, 10 articles were selected for analysis. A summary of the studies included is shown in Table 1 [3,7,8,9,10,11,12].

## 3. Results

In total, 10 studies with a total of 15 patients were investigated. In 10 cases (66.6%), the primary tumor was of serous histology and interestingly, in one case (6.66%), the primary tumor was a borderline tumor. By the time of the MM, 11 of the patients (73.3%) suffered from ovarian cancer recurrence, and surprisingly, the median time period between the initial diagnosis and the mediastinal recurrence was 13 years, while in 1 patient, mediastinal disease presented 20 months before the ovarian malignancy was diagnosed. Only in four patients (26.7%) was the mediastinal metastasis diagnosed simultaneously with the primary disease. The major symptoms of the patients were dyspnea in four cases (26.7%) and hemoptysis in one patient (6.66%). The median age at the time of diagnosis of the metastases was 58 years, including four patients at fertile age (26.6%). In total, 14 of the patients (93.3%) had undergone either primary or intermediate cytoreduction at the time of the diagnosis. The included studies [3,4,5,6,7,8,9,10,11,12] showed variations in the method of treatment since 10 patients (66.6%) underwent surgical resection, sometimes followed by adjuvant chemotherapy. More specifically, eight patients underwent transdiaphragmatic incision (53.3%), which was combined in one patient with the placement of a tracheobronchial stent, in another by rigid bronchoscopy, right thoracotomy, pleural biopsy, de-roofing of the mediastinal cyst and pleurodesis in the patient with hemoptysis, where the trachea was covered with a serratus anterior flap, combined with bronchoscopic laser debulking to open up the right mainstem bronchus and to control hemoptysis. Two patients (13.3%) underwent Video-Assisted Thoracic Surgery (VATS). No surgical complication was reported. Alternative management included chemotherapy in two patients (13.3%) and expectative management with observation in another three patients (20%), since the patients lacked any symptoms and impressively remained stable and asymptomatic during the follow-up period. The median follow-up was 11 months.

## 4. Discussion

It has been reported that the mediastinum could be the only site of metastasis when the primary tumor is in the ovaries with an incidence of up to 2.3%. [4]. The low incidence of mediastinal spread in ovarian carcinoma was attributed to locoregional dissemination mainly associated with advanced-stage disease [1]. On the other hand, Zannoni et. al. reported an interesting case of OC with mediastinal involvement as the first manifestation of the disease, 20 months before the abdominal mass could radiologically be identified [10].

Although mediastinal metastasis reports have shown favorable long-term results following repeated surgical resection, chemotherapy or even after surveillance and several reports of spreading bypassing the upper abdomen have been mentioned over the years, there has never been any postulation for the rationale associating ovarian cancer with isolated mediastinal metastases [13]. The behavior of ovarian carcinoma is distinct and differs significantly from the hematogenous metastasis commonly seen in other types of cancer. The majority of ovarian metastases are found within the peritoneal cavity. Once cancer cells have detached from the primary ovarian tumor, either as single cells or clusters, they metastasize passively through the movement of peritoneal fluid, invading the mesothelial cell layers of the peritoneum and omentum. Distribution outside of the peritoneum through the vasculature is less frequent in ovarian cancer (16%) and is typically associated with stage IV disease and a poor prognosis. The most common sites of distant metastasis are the liver (12.6%), pleura (6.6%), and lungs (4.6%) [11]. The precise mechanism accounting for this phenomenon of skip metastasis bypassing the first draining solid organ or the sentinel node through the hematogenous and lymphatics route is unclear but these features reinforced a heightened awareness at all times in detecting metastatic disease during the management of all ovarian, salpingeal and peritoneal malignancies.

When it comes to identification of such rare metastases, radiology appears to play an essential role, especially in asymptomatic patients. Typically, computer tomography (CT) is the preferable radiological method for OC patients’ surveillance and staging; however, in our study, three cases (20%) were initially demonstrated in a chest X-ray. In a case report by Oguchi et. al., the diagnosis was histologically confirmed following radiological evidence of mediastinal metastasis shown in single photon emission tomography (SPECT) and Tc-99m MDP (Technetium 99m-methyl diphosphonate) [9]. Another possible method to localize suspicious is utilising photon emission tomography/computer tomography (PET/CT), as described in the cases presented by Inoue et.al. [12] and Zannoni et al. [13]. Interestingly, some of the cases lacked histological confirmation and the therapeutic management was planned based on the radiological findings [8]. However, the present review supports the hypothesis that patients with ovarian cancers should be followed up with CT scans of the thorax for staging and surveillance routinely, since only two of the metastases were synchronous with the disease diagnosis [3,11], while the majority of the lesions in the present series occurred within several years after the initial presentation [7,8,9,12,13].

Curative treatment of late recurrence from ovarian cancer is not typically successful, although in certain cases of late recurrence from ovarian cancer, secondary debulking surgery may be recommended for selected patients. A retrospective cohort study involving 123 patients with recurrent ovarian cancer found that complete secondary cytoreduction had the greatest impact on overall survival throughout the treatment course [12]. This is especially true for cases where recurrent disease is localized, the patient has a good performance status, and they responded well to initial therapy, and usually, a combination of secondary debulking surgery (cytoreduction) and chemotherapy is recommended. In our study, surgical excision was successfully performed in 10 of the patients (66.6%), in 8 via thoracic surgery and in 2 via the minimal invasive technique of Video-Assisted Thoracic Surgery (VATS). Four of the patients did not receive any treatment (26.7%) [8], and in only one (6.7%) was palliative systemic therapy and immunotherapy administered [14], while Miura et al. recently reported a case of ovarian cancer mediastinal recurrence that was treated with a combination of radiotherapy and six cycles of carboplatin and paclitaxel after removing the metastatic tumor via median sternotomy and diaphragm resection [15].

The initial classification of FIGO stage IV ovarian cancer categorized patients based on the extent of the disease, prognosis, and recommended management. This included the presence of extra-peritoneal disease such as pleural effusion, liver metastases, and involvement of lymph nodes outside the abdomen, such as the supraclavicular lymph nodes. However, in the 2014 revision, patients with pleural effusion were considered as a separate category from those with parenchymal disease or metastases to extra-abdominal lymph nodes. Nasioudis et al. discovered that isolated distant lymph node metastases have a more favorable prognosis compared to stage IV disease with metastases in other sites, and are similar to those of patients with stage IIIC disease [16]. Zang et al. compared stage IV ovarian cancer patients with metastasis either outside the abdomen or in the liver and found that women with isolated supraclavicular lymphadenopathy or malignant pleural effusion had better survival rates than other stage IV patients [17]. Deng et al. conducted a study with 1481 patients and found that the site of distant metastases significantly impacts the overall prognosis in FIGO stage IV ovarian cancer patients. They observed a lower overall survival for parenchymal metastases compared to distant lymph node metastases [18]. Similarly, Herpje et al. found in a large study that women with stage IV serous ovarian cancer who only had lymph node involvement as distant metastasis lived longer than other stage IV patients [19]. A recent review by Pergaliotis et al. demonstrated that ovarian cancer patients with isolated lymph node recurrence have prolonged survival compared to recurrences in other sites as well. This type of recurrence appears to be less aggressive and can be treated with a combination of secondary cytoreduction and standard chemotherapy in selected cases [20]. This aligns with previous research conducted by Uzan et al., which suggested that patients with prior isolated ovarian cancer nodal recurrence may have a more favorable prognosis when undergoing surgical resection followed by chemoradiation or radiation therapy [21].

A basic limitation of our study is that isolated mediastinal metastasis is an absolute rarity of less clinical significance, since in any patient with ovarian cancer history where a suspicious mass is identified on a full body CT scan, a biopsy is required. Additionally, in cases of singular lesions, surgical removal would likely be offered to the patients alongside systemic therapy or radiotherapy. However, our study illuminated approaches where the patients received chemotherapy alone or expectant management with observation, with some cases lacking histological confirmation as well. Despite surgical dissection being a standard procedure that does not solely depend on the localization of the lesion, our review highlights that even though the mediastinum is not usually the first region of metastasis, excision of the mass can be a viable approach for certain subsets of ovarian cancer patients with isolated mediastinal metastasis (IMM), with the goal of improving survival rates. Our findings align with a recent publication from the Memorial Sloan Kettering (MSK) Cancer Center team, which demonstrated the safety and feasibility of intrathoracic cytoreduction in 178 advanced-stage ovarian cancer patients. Their study also suggested that this approach could lead to significantly improved progression-free survival (PFS) and overall survival (OS) in carefully selected patients who can undergo tumor-free surgery, despite having extra-abdominal tumor burden. Additionally, the same study recommends evaluating the eligibility of patients with operable stage IV disease for primary debulking surgery (PDS) instead of automatically considering neoadjuvant chemotherapy (NACT). Recent data indicate promising results when comparing NACT to PDS, suggesting that high-tumor-burden stage IV patients may not always require NACT as previously suggested in the classical approach [22]. However, further limitations include the small numbers of total cases and the use of retrospective data to acquire reports of cases of prostration. Another limitation is that our systematic review was not registered in PROSPERO; therefore, it can be crosschecked that the study was conducted as planned. Last, this is a self-funded study and due to limited access to other databases such as Embase and Scopus, only three databases (PubMed, Google and Cochrane Library) were included, thus representing a slight selection bias.

Hence, our study draws attention to an almost neglected disease entity of ovarian cancer, namely isolated mediastinal metastases. In the light of the existing literature, the different options in the treatment outcomes are presented, outspreading from surgical removal either via transdiaphragmatic incision in the majority of the cases or Video-Assisted Thoracic Surgery (VATS) to the conservative approach with chemotherapy or even expectative management with observation. The variety in the therapeutic methods is indicative of the need of highly specialized care of such rare cases and is giving credence to handling those patients in a multidisciplinary team and at tertiary centers, where highly selected patients eligible for surgical treatment might undergo curative intent cytoreduction with a fair chance of long-term disease control. Individual therapy planning is also required for patients with a desire to become pregnant, although based on the current recommendations, fertility-sparing surgery with preservation of the uterus with or without preservation of the contralateral annex is only acceptable in young patients with early-stage ovarian carcinoma and it should not be considered in patients with advanced disease [23].

## 5. Conclusions

As shown by the size of the present series and literature review, isolated mediastinal metastases with a very low incidence lack a specific treatment strategy, while systemic therapy is typically administered, based on the general recommendations for OC stage IV disease. The true incidence of isolated mediastinal metastases without abdominal metastases in ovarian cancer could be low, but this remains unclear. Nevertheless, our findings highlight the importance of including the search for mediastinal disease in the staging and surveillance stages of all patients with ovarian cancer and practically recommending to the clinician to not preclude a search for mediastinal metastases in the absence of abdominal involvement. Furthermore, although the current trend is to administer chemotherapy in patients with stage IV disease, exploring the potential feasibility of optimal disease debulking remains controversial. Our review provides a comprehensive summary of the clinical characteristics, treatment methods, and outcomes of MM in ovarian cancer patients, which is to our knowledge not extensively covered in other literature. By compiling data from 15 reported cases, our study highlights the diversity in treatment approaches and the potential for surgical intervention, adding to the body of knowledge on managing advanced-stage ovarian cancer. Multi-disciplinary team meetings during which these cases are discussed are essential to outline an optimal strategy and evaluate the option of surgical cytoreduction. Last but not least, since anterior mediastinal metastasis from OC can cause major complications, future research in the management of isolated mediastinal metastasis in a case of ovarian cancer is needed and a large number of cases must be accumulated to establish optimal management, including more detailed patient data and standardizing the outcome measures across cases to improve the robustness of the findings.

## Figures and Tables

**Figure 1 life-14-01098-f001:**
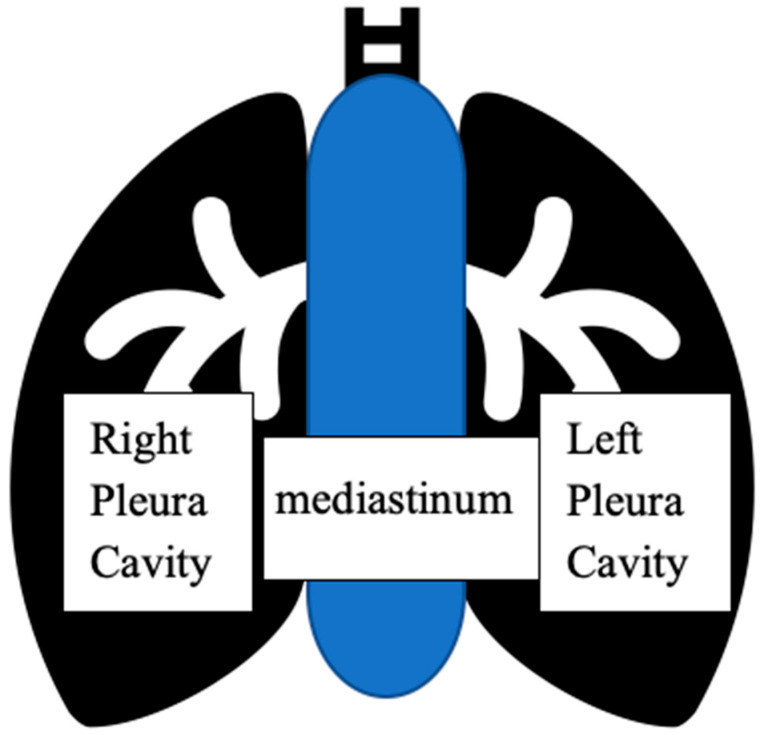
The mediastinum located between the pleural cavities, which enclose the lungs.

**Figure 2 life-14-01098-f002:**
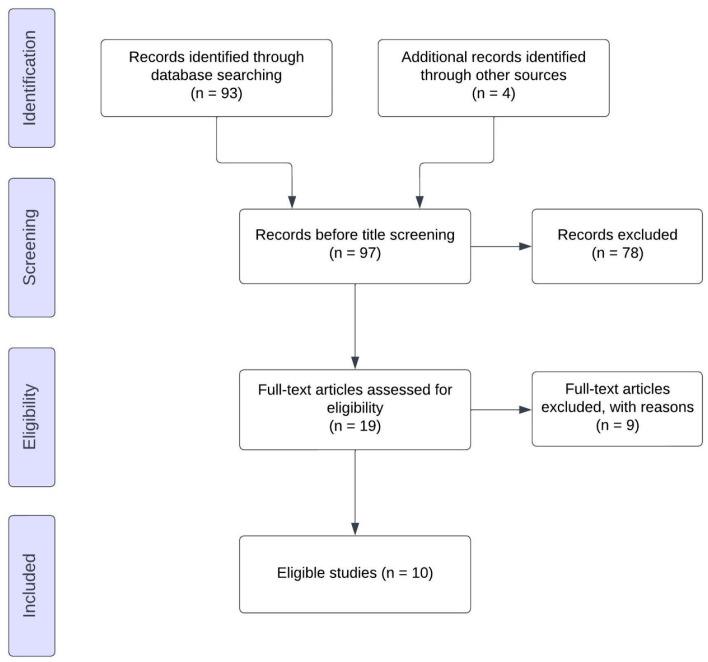
PRISMA flow diagram.

**Table 1 life-14-01098-t001:** Studies reporting data on isolated mediastinal metastasis in ovarian cancer patients.

First Author Year,	Nr of Patients	Age of Patients (Range)	Serous Histology	Other Histology	Primary Disease	Recurrence	Time Period from Initial Diagnosis to Recurrence (Years)	Main Symptoms	Primary Disease Treatment		Mediastinal Resection	LN Treated with Systematic Therapy	Follow-Up (mo)
									Chemo Therapy	Surgical Debulking			
Case series													
Moran, 2005 [10]	3	Median: 40 (33–50)	3	0	0	3	NM	None	0	3	3	0	Median: 12 (6–18)
Blanchard, 2007 [8]	4	Median: 62.5 (58–67)	NM	NM	0	4	4.5	None	0	4	0	0	Median: 75.5 (19–132)
Case reports													
Oguchi, 1998 [9]	1	41	1	0	0	1		Dyspnea	0	1	1	0	5
Montero, 2000 [11]	1	46	1	0	0	1	0	None	0	1	1 (VATS)	0	6
Zannoni, 2006 [13]	1	63	1	0	1	0	20 months earlier	Dyspnea	1 (NACT)	1	1	0	1.5
Scarci, 2010 [3]	1	71	1	0	1	0		Dyspnea	1 (NACT)	1	0	1	10
Dhilon, 2016 [7]	1	61	1	0	0	1		Bilateral mass	0	1	1	0	36
Inoue, 2020 [12]	1	67	1	0	0	1	29	None	1	1	1 (VATS)	0	12
Solek, 2021 [14]	1	48	0	1	0	1		Dyspnea	0	1	0	1	7
Miura, 2022 [15]	1	88	1	0	0	1	16	None	1	1	1	0	10

Abbreviations: NM = not mentioned.

## Data Availability

Publicly available datasets were analyzed in this study. This data can be found here: https://pubmed.ncbi.nlm.nih.gov/; https://scholar.google.com/; https://www.cochranelibrary.com/.

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
