# Peer review of "Mediastinal Metastasis Isolated in Ovarian Cancer: A Systematic Review"

_life, 2024, doi:10.3390/life14091098_

Round 1

Reviewer 1 Report (New Reviewer)

Comments and Suggestions for Authors

In their work, the authors have highlighted a very interesting and important topic, although, as the authors point out, it is not very common one, but perhaps because it is quite neglected. The paper is an overview and a review of the available literature. However, it is worth expanding precisely the background of this work by expanding and updating the data and literature presented. Even though this is not a topic often covered, it may be worthwhile to analyze other sources as well. The work also needs to be rearranged in terms of the graphic elements included at the end of the work, which should be in their respective sections. In conclusion, the topic raised by the authors is certainly valuable from the point of view of drawing the attention of clinicians, especially. However, it needs refinement.   

Comments on the Quality of English Language

No comment

Author Response

Comment 1: In their work, the authors have highlighted a very interesting and important topic, although, as the authors point out, it is not very common one, but perhaps because it is quite neglected. The paper is an overview and a review of the available literature. However, it is worth expanding precisely the background of this work by expanding and updating the data and literature presented. Even though this is not a topic often covered, it may be worthwhile to analyze other sources as well. The work also needs to be rearranged in terms of the graphic elements included at the end of the work, which should be in their respective sections. In conclusion, the topic raised by the authors is certainly valuable from the point of view of drawing the attention of clinicians, especially. However, it needs refinement.   

Response 1: We would like to thank the reviewer for their valuable comments and their helpful report! We updated the data presented and regarding the literature expansion we commented in our limitation paragraph the following:

“Last, this is a self-funded study and due to limited access in other databases such as Embase and Scopus only two three databases (PubMed,Google and Cochrane Library) were included, thus representing a slight selection bias.

Reviewer 2 Report (New Reviewer)

Comments and Suggestions for Authors

The manuscript entitled "Mediastinal Metastasis Isolated in Ovarian Cancer: A Systematic Review" is written in clearly understandable english and covers  cases of mediastinal metastasis in case of ovarian cancer. These cases are very rare. In Table 1 authors mention 10 works describing these cases. 

I have some recommendations for authors for improvig the quality of the manuscript:

1. Provide some figure illustrating what is mediastinal and how metastasis from ovary appears there.

2. Provide full meaning of abbreviations (line 100 - "MM". Line 153 "CT", Line 159 "PET").

3. Typeset line 156 "wad"

4. How many women at fertile age suffered from such metastasis? Provide brief discussion if is it required some procedures for fertility preservation? Which kind of fertility preservation for such patients seems to be better?

Author Response

Comment 2:

The manuscript entitled "Mediastinal Metastasis Isolated in Ovarian Cancer: A Systematic Review" is written in clearly understandable english and covers  cases of mediastinal metastasis in case of ovarian cancer. These cases are very rare. In Table 1 authors mention 10 works describing these cases. 

I have some recommendations for authors for improvig the quality of the manuscript:

1. Provide some figure illustrating what is mediastinal and how metastasis from ovary appears there.

2. Provide full meaning of abbreviations (line 100 - "MM". Line 153 "CT", Line 159 "PET").

3. Typeset line 156 "wad"

4. How many women at fertile age suffered from such metastasis? Provide brief discussion if is it required some procedures for fertility preservation? Which kind of fertility preservation for such patients seems to be better?

Response 2: We thank the reviewer for the comments! We have, accordingly revised our manuscript to emphasize the points.

  1. We provide an illustrative figure of mediastinum.
  2. We provided the full meaning of the abbreviations, thank you for pointing that out.
  3. We corrected the typeset.
  4. We would like to thank again the reviewer for their detailed comments. We, have, accordingly modified the Discussion part as follows:

Individual therapy plan is also required for patients with a desire to become pregnant, although based on the current recommendations, fertility-sparing surgery with preservation of the uterus with or without preservation of the contralateral annex, is only acceptable in young patients with early stage ovarian carcinoma and it should not be considered in patients with advanced disease

Reviewer 3 Report (New Reviewer)

Comments and Suggestions for Authors

The main question addressed by the research is how isolated mediastinal metastasis in ovarian cancer is managed and what the outcomes are for patients undergoing various treatment approaches.

The topic is original as isolated mediastinal metastasis in ovarian cancer is exceedingly rare and not widely studied. 

The study is relevant but it would be more useful by including in the introduction 1-2 parragraphs that demonstrate the value of a multidisciplinary approach and individualized treatment in managing rare gynecological cancer cases that align with the conclusions of the current review (e.g. Dueño S, Stein R, Jamal M, Lewis G, Hew K. Metastasis of serous ovarian carcinoma to the breast: a case report and review of the literature. J Med Case Rep. 2024;18(1):127. Published 2024 Mar 26. doi:10.1186/s13256-024-04445-y; Georgescu, M.T.; Georgescu, D.E.; Georgescu, T.F.; Serbanescu, L.G. Changing the Prognosis of Metastatic Cervix Uteri Adenosquamous Carcinoma through a Multimodal Approach: A Case Report. Case Rep. Oncol. 2020, 13, 1545–1551 and Chan, J. K., Chow, S., Bhowmik, S., Mann, A., Kapp, D. S., & Coleman, R. L. (2018). Metastatic gynecologic malignancies: advances in treatment and management. Clinical & experimental metastasis35(5-6), 521–533. https://doi.org/10.1007/s10585-018-9889-7) . My recommendation would be to improve the refferences list with  manuscripts like the ones previously suggested that would further emphasize the importance of personalized and innovative treatment strategies in rare metastatic conditions.

This systematic review provides a comprehensive summary of the clinical characteristics, treatment methods, and outcomes of IMM in ovarian cancer patients, which is not extensively covered in other literature. By compiling data from 15 reported cases, the study highlights the diversity in treatment approaches and the potential for surgical intervention, adding to the body of knowledge on managing advanced-stage ovarian cancer.

An importatn drawback of this study would be that the review relies on only two databases (PubMed and Google), which may introduce selection bias. Expanding the search to include other databases such as Embase, Scopus, and Cochrane Library would provide a more comprehensive literature base.

Including more detailed patient data and standardizing the outcome measures across cases could improve the robustness of the findings.

The conclusions  address the main question posed by highlighting the rarity of IMM and the various management strategies used. However, the small number of cases and the retrospective nature of the data limit the ability to draw definitive conclusions.

The references are appropriate and relevant to the topic. This list should further be expnaded to include a mix of case reports and studies that provide context and support for the findings.

Author Response

Response 2: We thank the reviewer for the comments! We have, accordingly revised our manuscript to emphasize the points.

1.     We provide an illustrative figure of mediastinum.

2.     We provided the full meaning of the abbreviations, thank you for pointing that out.

3.     We corrected the typeset.

4.     We would like to thank again the reviewer for their detailed comments. We, have, accordingly modified the Discussion part as follows:

Individual therapy plan is also required for patients with a desire to become pregnant, although based on the current recommendations, fertility-sparing surgery with preservation of the uterus with or without preservation of the contralateral annex, is only acceptable in young patients with early stage ovarian carcinoma and it should not be considered in patients with advanced disease

Comment 3:

The main question addressed by the research is how isolated mediastinal metastasis in ovarian cancer is managed and what the outcomes are for patients undergoing various treatment approaches.

The topic is original as isolated mediastinal metastasis in ovarian cancer is exceedingly rare and not widely studied. 

The study is relevant but it would be more useful by including in the introduction 1-2 parragraphs that demonstrate the value of a multidisciplinary approach and individualized treatment in managing rare gynecological cancer cases that align with the conclusions of the current review (e.g. Dueño S, Stein R, Jamal M, Lewis G, Hew K. Metastasis of serous ovarian carcinoma to the breast: a case report and review of the literature. J Med Case Rep. 2024;18(1):127. Published 2024 Mar 26. doi:10.1186/s13256-024-04445-y; Georgescu, M.T.; Georgescu, D.E.; Georgescu, T.F.; Serbanescu, L.G. Changing the Prognosis of Metastatic Cervix Uteri Adenosquamous Carcinoma through a Multimodal Approach: A Case Report. Case Rep. Oncol. 2020, 13, 1545–1551 and Chan, J. K., Chow, S., Bhowmik, S., Mann, A., Kapp, D. S., & Coleman, R. L. (2018). Metastatic gynecologic malignancies: advances in treatment and management. Clinical & experimental metastasis35(5-6), 521–533. https://doi.org/10.1007/s10585-018-9889-7) . My recommendation would be to improve the refferences list with manuscripts like the ones previously suggested that would further emphasize the importance of personalized and innovative treatment strategies in rare metastatic conditions.

This systematic review provides a comprehensive summary of the clinical characteristics, treatment methods, and outcomes of IMM in ovarian cancer patients, which is not extensively covered in other literature. By compiling data from 15 reported cases, the study highlights the diversity in treatment approaches and the potential for surgical intervention, adding to the body of knowledge on managing advanced-stage ovarian cancer.

An importatn drawback of this study would be that the review relies on only two databases (PubMed and Google), which may introduce selection bias. Expanding the search to include other databases such as Embase, Scopus, and Cochrane Library would provide a more comprehensive literature base.

Including more detailed patient data and standardizing the outcome measures across cases could improve the robustness of the findings.

The conclusions  address the main question posed by highlighting the rarity of IMM and the various management strategies used. However, the small number of cases and the retrospective nature of the data limit the ability to draw definitive conclusions.

The references are appropriate and relevant to the topic. This list should further be expnaded to include a mix of case reports and studies that provide context and support for the findings.

Response 3: We agree with the comment and thank the reviewer for suggesting relative manuscripts. We have modified the introduction accordingly:

However, a substantial number of cases need to be gathered to determine the best management strategy, and a multidisciplinary approach and individualized treatment are always crucial in managing rare gynecological cancer cases, since the importance of personalized and innovative treatment strategies in rare metastatic conditions has been emphasized in many studies, suggesting that radiotherapy or neoadjuvant chemotherapy could also be considered [4-6]. 

Regarding the databases searched we expanded our search including Cochrane Library and commented on the possible selection bias in our limitation paragraph. We also referred to the small number of cases and the retrospective nature of the data as another study limitation.

Round 2

Reviewer 3 Report (New Reviewer)

Comments and Suggestions for Authors

I agree with the publication of the manuscript in the current form

This manuscript is a resubmission of an earlier submission. The following is a list of the peer review reports and author responses from that submission.

Round 1

Reviewer 1 Report

Comments and Suggestions for Authors

The described topic is an absolute rarity of low clinical significance that does not offer any novel insights in the context of ovarian cancer diagnosis or treatment. In case a suspicious mass is identified on a full body CT scan (mediastinal or other regions) in a patient with ovarian cancer history, a biopsy is required, and in case of singular lesions, surgical removal will most likely be offered to the patients alongside with systemic therapy or radiotherapy. This is a standard procedure and does not solely depend on the localization of the lesion, even though the mediastinum is not usually the first region of metastasis.

Comments on the Quality of English Language

Minor editing

Reviewer 2 Report

Comments and Suggestions for Authors

Dear authors,

I have reviewed the manuscript "Ovarian Cancer and Isolated Mediastinal Metastasis: A Systematic Review." This document portrays a very important case, though relatively uncommon: isolated mediastinal metastasis in an ovarian malignancy. However, the methodology is incorrect and does not support the conclusion.

Below, I provide detailed comments with suggestions for improvement about the individual sections in the order of the paper.

Title

1)Specificity and Clarity: Your title is specific and very clear; however, it is also somewhat lengthy. Below are several suggestions to make it more concise: "Mediastinal Metastasis Isolated in Ovarian Cancer: A Systematic Review."

2) Abstract

result section needs to reflect the principal findings and conclusions of the study accurately. 

3) Introduction

Correct background: This is a good overview, but more information on the clinical implications of solitary mediastinal metastasis of ovarian cancers would be helpful.

4) Objective Clarity: Clearly describe that the major purpose of this paper is to identify related needs and explore feasibility and reveal outcomes associated with different treatment approaches.

5) Methods

Details of Search Strategy: Outline all elements of the search strategy undertaken, including full details of the search terms and how the search was constructed within each database. Explain in depth any limits initially set to searching and dates when done. The PRISMA flow diagram is provided here. Please describe in greater detail in the text below what you did for searching.

6) Inclusion and exclusion criteria: describe these characteristics and how studies were identified for more detailed review, or the reason for their rejection.

7) This systematic review was not registered in PROSPERO, and the protocol was not available. Therefore, it would be impossible to judge whether the review was conducted as planned. This represents another selection bias.

8) Why do the authors include only two databases (PubMed and Google)?  This represents another selection bias.

9) Why do the authors exclude non-english articles? This represents another selection bias.

10) Results

Characteristics of included studies A descriptive summary of the characteristics of included studies will be provided in tables along with information on population demographic data for patients and basic disease characteristics, such as the model of treatment.

11) Data Presentation: All data should be clearly presented and consistently. For example, the median follow-up time needs to be described better; also, there needs to be a definition of its range.

12) Discussion

Interpretation of findings: it should try to interpret the findings more critically in the light of existing literature and must discuss possible reasons for likely variations in the treatment outcomes, thereby giving credence to handling such cases in a multidisciplinary team.

13) Clinical Implications: Discuss the clinical implications of your findings, particularly how your findings will apply to influence practice and future research in the management of isolated mediastinal metastasis in a case of ovarian cancer.

14) Limitations: Clearly state the study's limitations, including small numbers of cases and the use of retrospective data to acquire reports of cases of prostration.

15) Conclusions

Conclusion Present: The conclusion would have to summarize the key points briefly and, if possible, it would be best to conclude by making practical recommendations to the clinician. Also, indication of future research needs and areas where further study can be based are needed.

Conclusion 

This manuscript presents numerous biases and can't be published in this form. 

Kind regards